# The effects of mycobacterial RmlA perturbation on cellular dNTP pool, cell morphology, and replication stress in *Mycobacterium smegmatis*

Rita Hirmondó[1]*, Ármin Horváth[1], Dániel Molnár[1,2], György Török[1,3], Liem Nguyen[4], Judit Tóth[1,5]*

1 Institute of Enzymology, Research Centre for Natural Sciences, Budapest, Hungary, 2 Doctoral School of Biology and Institute of Biology, Eötvös Loránd University, Budapest, Hungary, 3 Department of Biophysics and Radiation Biology, Semmelweis University, Budapest, Hungary, 4 Department of Medicine, Case Western Reserve University School of Medicine, Cleveland, Ohio, United States of America, 5 Department of Applied Biotechnology and Food Sciences, Budapest University of Technology and Economics, Budapest, Hungary

* hirmondo.rita@ttk.hu (RH); toth.judit@ttk.hu (JT)

**Data Availability Statement:** All relevant data are within the manuscript and its Supporting information files.

## Abstract

The concerted action of DNA replication and cell division has been extensively investigated in eukaryotes. Well demarcated checkpoints have been identified in the cell cycle, which provides the correct DNA stoichiometry and appropriate growth in the progeny. In bacteria, which grow faster and less concerted than eukaryotes, the linkages between cell elongation and DNA synthesis are unclear. dTTP, one of the canonical nucleotide-building blocks of DNA, is also used for cell wall biosynthesis in mycobacteria. We hypothesize that the interconnection between DNA and cell wall biosynthesis through dTTP may require synchronization of these processes by regulating dTTP availability. We investigated growth, morphology, cellular dNTP pool, and possible signs of stress in *Mycobacterium smegmatis* upon perturbation of rhamnose biosynthesis by the overexpression of RmlA. RmlA is a cell wall synthetic enzyme that uses dTTP as the precursor for cross-linking the peptidoglycan with the arabinogalactan layers by a phosphodiester bond in the mycobacterial cell wall. We found that RmlA overexpression results in changes in cell morphology, causing cell elongation and disruption of the cylindrical cell shape. We also found that the cellular dTTP pool is reduced by half in RmlA overexpressing cells and that this reduced dTTP availability does not restrict cell growth. We observed 2-6-fold increases in the gene expression of replication and cell wall biosynthesis stress factors upon RmlA overexpression. Using super-resolution microscopy, we found that RmlA, acting to crosslink the nascent layers of the cell wall, localizes throughout the whole cell length in a helical pattern in addition to the cellular pole.

**Funding:** This work was supported by the National Research, Development and Innovation Office, Hungary, OTKA FK124527 to JT and PD128254 to RH. Work in L.N. laboratory was supported by National Institutes of Health grants R01AI087903, R21AI119287 and R21AI159770. The funders had no role in study design, data collection and analysis, decision to publish, or preparation of the manuscript.

**Competing interests:** The authors have declared that no competing interests exist.

## Introduction

The unique intricate cell wall impermeable for most antibiotics is a specific hallmark of mycobacteria. This cell wall makes infections caused by pathogenic mycobacteria extremely difficult to treat [1, 2]. *Mycobacterium tuberculosis* (*M. tuberculosis*) has been a leading cause of mortality worldwide, accounting for 1.4 million deaths in 2020, commensurable only with the death toll caused by SARS-CoV-2 [3]. Although effective combination chemotherapy exists to combat *M. tuberculosis*, drug-resistant tuberculosis accounts for 450,000 new cases annually. Almost a third of all tuberculosis-related deaths are due to antimicrobial resistance [4]. The thick cell wall constitutes of mycolic acid, peptidoglycan, and arabinogalactan (Fig 1). The galactan region of the arabinogalactan layer is bound to the peptidoglycan layer via a phosphodiester linkage of the α-L-rhamnopyranose-(1→3)-α-D-GlcNAc-1-phosphate disaccharide [5]. This linker, composed of a rhamnosyl residue, a sugar not found in humans, is critical to the structural integrity of the mycobacterial cell wall (Fig 1), as well as for the viability and pathogenicity of pathogenic mycobacteria [6, 7]. Rhamnose has also been reported to mediate virulence, adhesion, and pathogenesis in several other bacteria [8–11]. L-Rhamnosyl residues are synthesized in the rhamnose pathway by four enzymes: RmlA, B, C, and D from α-D-glucose-1-phosphate (G1P) and dTTP as the main precursors [7]. No salvage pathway is known for the synthesis of dTDP-L-rhamnose. Consistent with this, the RmlA enzyme (D-glucose-1-phosphate thymidylyltransferase), which catalyzes the first step of rhamnose biosynthesis, is essential for bacterial growth [12]. The importance of this enzyme for mycobacterial viability, together with the absence of the rhamnose pathway in humans, make this enzyme a potentially valid drug target for the development of RmlA-related anti-mycobacterial drugs for tuberculosis treatment [13].

RmlA catalyzes the condensation of G1P with dTTP to produce dTDP-D-glucose, which is processed further by the RmlB-D enzymes [7]. RmlA is homologous to other bacterial sugar nucleotide transferases, although the tetrameric arrangement of RmlA is distinct from these [14, 15]. The competitive and non-competitive product inhibition of RmlA by dTDP-L-rhamnose has long been known [16]. In addition, the *M. tuberculosis* rhamnose pathway is likely regulated by c-di-GMP and the serine/threonine protein kinases PknG and PknB [13, 17]. PknG is required for the intrinsic multidrug resistance [18] and virulence [19] of mycobacteria; moreover, it interacts with and phosphorylates two essential components of the rhamnose pathway, RmlA and Wbbl2 [17], thereby inhibiting the enzymatic activity of RmlA [17]. The inhibition of cell wall assembly has proven to be effective for inhibiting mycobacterial growth. Drugs such as ethambutol (EMB), isoniazid (INH), and D-cycloserine, which target the synthesis of various cell wall components, are successfully used in tuberculosis therapy. As a result, the biosynthesis of the mycobacterial cell wall has been a significant research objective over the last decade [1, 5, 20].

As dTTP, one of the canonical nucleotide building blocks of DNA, is utilized not only for DNA replication but also for cell wall biosynthesis and assembly, we hypothesized that this connection might serve as a checkpoint allowing mycobacteria to synchronize these processes by monitoring and regulating dTTP availability (Fig 1). To this end, we investigated growth, morphology, cellular dNTP pool, and possible signs of stress in *Mycobacterium smegmatis* (*M. smegmatis*) upon perturbation of the cellular RmlA level. Since *rmlA* is essential [12] and attempts to inhibit *M. smegmatis* RmlA using a previously reported inhibitor [21] were unsuccessful, we chose to overexpress RmlA to influence and study its cellular function. In addition, by overexpressing RmlA, cellular dTTP used for replication could be restricted. We found that RmlA overexpression caused morphological changes, including cell elongation and the appearance of large polar bulbs at the tip of the cell. RmlA overexpression induced 2-6-fold changes in the mRNA levels of various stress factors, including LexA, WhmD, and IniA. We

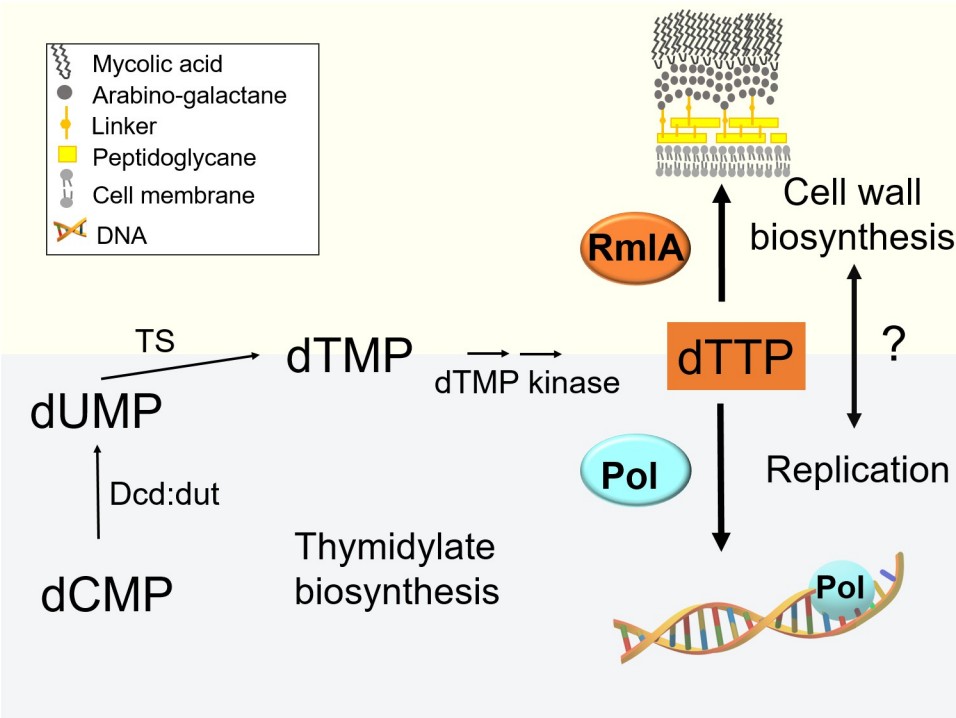

**Fig 1. Hypothesis: A dual role of thymidylate biosynthesis in mycobacterium.** In mycobacteria, dUMP, the precursor for dTTP biosynthesis, is synthesized exclusively by dCTP deamination and coupled dUTP hydrolysis by Dcd:dut. dTTP is used not only for DNA replication by DNA polymerases but also by RmlA that uses this nucleotide for the first step of L-rhamnosyl biosynthesis, a critical step in cell wall biosynthesis. (Abbreviations: linker–L-rhamnosyl linker; Dcd:dut—bifunctional dCTP deaminase:dUTPase; TS–thymidylate synthase; RmlA—D-glucose-1-phosphate thymidylyltransferase; Pol–DNA polymerase).

also found that RmlA overexpression resulted in significant dTTP depletion in a *M. smegmatis* PknG knock-out strain (PknG(-)). Finally, live-cell imaging of *M. smegmatis* cells expressing fluorescently tagged RmlA revealed an intriguing helix-like distribution of RmlA along the cylindrical portion of the cell.

## Materials and methods

### Bacterial strains, media, and growth conditions

*M. smegmatis* mc$^2$155 [22] strains were grown in Lemco broth. The Lemco broth was supplemented with 15 g L−1 Bacto agar for a solid medium. Streptomycin was added at 20 μg/ml, hygromycin B at 50 μg/ml, and tetracycline (TC) at the final concentration of 0.05–0.2 μg/ml.

### Construction of RmlA expressing *M. smegmatis* strains

The RmlA coding region was PCR amplified and cloned into the vectors pKW08-Lx [23] (Addgene #25012), pTEC16-mOrange [24] (Addgene plasmid # 30175), and pLL192 [25] using the primers and restriction sites indicated in S1 Table in S1 File to construct the overexpressing and the fluorescently tagged constructs, respectively. Successful cloning was verified with the sequencing of the appropriate region of the plasmid. 0.5–0.5 μg of the appropriate plasmids were electroporated into electrocompetent wild type (WT) [26] or PknG(-) [18] *M. smegmatis* strain according to Table 1. Three parallels of each strain were used in the experiments.

**Table 1. Strains and plasmids used in the study.**

| | resistance | relevant characteristics | usage | source or reference |
|---|---|---|---|---|
| **strains** | | | | |
| mc2 155 | - | WT | | Snapper et al. 1990 |
| RmlAi | strep, hyg | wt harboring pKW08-RmlA | inducible (TC) overexpression of RmlA | this study |
| RmlAc | hyg | wt harboring pTEC-RmlAover | constant overexpression of RmlA | this study |
| Lxi | hyg | wt harboring pKW08-Lx | inducible (TC) overexpression of luciferase, control strain | Hirmondo et al. 2015 |
| RmlA-gfp | strep | wt harboring pLL192-RmlA | expression of RmlA-gfp | this study |
| RmlA-mOrange | hyg | wt harboring pTEC16-RmlA | expression of RmlA-mOrange-2 | this study |
| mOrange | hyg | wt harboring pTEC16 | expression of mOrange-2; control strain | this study |
| PknG- | hyg | pknG- | control strain | Wolff et al. 2009 |
| RmlAi, pknG- | strep, hyg | pknG- harboring pKW08-RmlA | inducible (TC) overexpression of RmlA in PknG-background | this study |
| **plasmids** | | | | |
| pKW08-Lx | hyg | expression from TetRO promoter | inducible (TC) overexpression of luciferase | Williams et al. 2010; Addgene #25012 |
| pKW08-RmlA | hyg, strep | expression from TetRO promoter | inducible (TC) overexpression of RmlA | this study |
| pTEC-RmlAover | hyg | expression from hsp60 promoter | constant overexpression of RmlA | this study |
| pLL192-RmlA | strep | GFP tag, expression from own promoter | localization of RmlA | this study |
| pTEC16 | hyg | mOrange-2 expression from MSP promoter | localization control | Takaki et al. 2013; Addgene #30175 |
| pTEC16-RmlA | hyg | mOrange-2 tag, expression from own promoter | localization of RmlA; super-resolution imaging | this study |

## Verification of RmlA expression by qPCR

Overexpression was verified by measuring mRNA levels of RmlA in WT and inducible or constitutive overexpressing strains. Cells were grown in 50 ml liquid culture until saturation; then, in the RmlAi strain, overexpression was induced by adding 0.05 μg/ml TC at OD(600) = 0.2–0.4. Cells were harvested by centrifugation at 4000 g for 20 min. Cells were washed in ice-cold PBS before resuspension in 2 volumes of RNA Protect Reagent (Qiagen). Samples were stored at -80˚C. RNA was purified with the Rneasy RNA Clean-up kit (Qiagen) according to the manufacturer's instructions. Mycobacterial RNA yield was assayed using the Nano-Drop ND-2000 Spectrophotometer (NanoDrop Technologies). cDNA samples were amplified from 0.1 μg total RNA by random hexamer primers using High-Capacity Reverse Transcription Kit (Applied Bioscience). The resulting cDNA was quantified by qPCR using EvaGreen (Bioline) and MyTaq PCR master mix (Bioline) in a BioRad CFX96 qPCR instrument. Non-reverse transcribed, and no template controls were used as controls for DNA contamination. We prepared three technical and three biological replicates for all measurements. *sigA* (MSMEG_2758) and Ffh (MSMEG_2430) [27] were used as endogenous reference genes to normalize input cDNA concentration. The relative expression ratios of the examined genes were calculated using the comparative Ct method (ΔΔCt) by the BioRad CFX Maestro software. Primers, primer efficiency, and all measured data of the qPCR analysis are compiled in the supplementary archive.

## Growth inhibition assay

The control WT and PknG(-) *M. smegmatis* strains, the WT and PknG(-) *M. smegmatis* strains overexpressing RmlA (RmlAi), and the control *M. smegmatis* strain overexpressing luciferase (Lxi) were grown in M9 minimal media containing 0.05% Tween-80 in liquid culture overnight. The precultures were then diluted to OD (600) = 0.1 and grown in the presence of

various concentrations of TC (0, 0.05, or 0.1 µg/ml TC) in a plate reader (Biotek Synergy MX) at 37˚C with constant shaking. OD (600) was measured every 10 min.

## Microscopy

RmlA was overexpressed in WT and PknG(-) strains for morphological studies. Overexpression was induced by 0.1 or 0.2 µg/ml TC. The strains were grown in agar plates containing TC in the indicated concentration, then streaked onto microscopy slides covered with 0.1% low melting agarose (Sigma). Cell membranes were stained with 1 µg/ml Bodipy (522/529) dye (Thermo Fischer). Samples were investigated in phase-contrast, and fluorescence modes under a Leica DM IL LED microscope. Microscopic images were analyzed using Fiji [28]. Cell length was measured using the software Bacstalk [29].

RmlA was expressed using its own promoter for localization studies, tagged with GFP or mOrange-2. If appropriate, the strains were grown in agar plates containing specific drugs, then streaked onto microscopy slides covered with 0.1% low melting agarose (Sigma). RmlA-GFP strains were stained with 10 µg/ml Propidium-iodide. Samples were investigated using a Leica TCS SP8 STED microscope. In the case of the GFP tagged constructs, the confocal mode was used, while the mOrange-2 signal was suitable for the STED mode to analyze localization patterns. Deconvolution was performed using the Huygens Professional software. Microscopic images were refined and analyzed using Fiji [28].

## Gene expression analysis

RmlA was overexpressed in WT and PknG(-) strains for gene expression studies. Overexpression was induced by 0.1 or 0.2 µg/ml TC. For CIP and EMB treatments, drugs were added to the cultures at OD (600) = 0.1. CIP was used at 0.3 µg/ml, EMB at 100 µg/ml final concentration, where cell mortality was 20–80%. Cells were grown in 50 ml liquid culture until saturation, washed in ice-cold PBS, and harvested by centrifugation (4000 g, 20 min). Bacterial pellets were resuspended in 2 volumes of RNA Protect Reagent (Qiagen) and stored at -80˚C. RNA was purified with the Rneasy RNA Clean-up kit (Qiagen) according to the manufacturer's instructions. Mycobacterial RNA yields were assayed using the Nano-Drop ND-2000 Spectrophotometer (Nano-Drop Technologies). 0.1 µg of total RNA was reverse transcribed to cDNA using random hexamer primers and the High-Capacity Reverse Transcription Kit (Applied Bioscience). The resulting cDNA was quantified by qPCR using EvaGreen (Bioline) and MyTaq PCR master mix (Bioline) in a BioRad CFX96 qPCR instrument. Non-reverse transcribed, and no-template controls were used for checking genomic and exogenous DNA contamination, respectively. For all measurements, three technical and three biological replicates were used. *sigA* (MSMEG_2758) and Ffh (MSMEG_2430) were used as endogenous reference genes to normalize input cDNA concentrations. The relative expression ratios of the examined genes were calculated using the comparative Ct method (ΔΔCt) by the BioRad CFX Maestro software. Primers, primer efficiency, and all measured data of the qPCR analysis are compiled in S1 Table in S1 File.

## dNTP extraction and determination of the dNTP pool size

dNTP extraction and quantification were performed according to Szabo et al. [30]. Cells were grown until the culture reached the mid-exponential phase OD (600) ~ 0.7. The total CFUs were determined for each culture. The cultures were centrifuged (20 min, 4000 g, 4˚C), and the cell pellets were extracted in precooled 0.5 ml 60% methanol overnight at −20˚C. After 5 minutes of boiling at 95˚C, cell debris was removed by centrifugation (20 min, 13 400 g, 4˚C). The methanolic supernatant containing the soluble dNTP fraction was vacuum-dried (Eppendorf) at 45˚C. The dNTP pellet was dissolved in 50 µl nuclease-free water.

Determination of the dNTP pool size in each extract was as follows: 10 pmol template oligo (Sigma), 10 pmol probe (IDT), and 10 pmol NDP1 primers (Sigma) were present per 25 μl reaction. The concentration of each non-specific dNTP was kept at 100 μM. VWR® TEMPase Hot Start DNA Polymerase (VWR) was used at 0.9 unit/reaction in the presence of 2.5 mM MgCl$_2$. To record calibration curves, the reaction was supplied with 0–12 pmol or 0–25 pmol specific dNTP depending on the applied dT1 or dT2 template, respectively. Sequences of used primers and probes are presented in the supplementary archive. Fluorescence was recorded every 13 seconds in a CFX96 Touch™ Real-Time PCR Detection System. The thermal profile was 95˚C 15 min, (60˚C 13 s) × 260 cycle for dATP and dTTP measurement. In the case of dCTP and dGTP measurements, the polymerization temperature was 55˚C. The results were analyzed using the nucleoTIDY software (http://nucleotidy.enzim.ttk.mta.hu/) [30].

## Results

### RmlA overexpression produces elongated cells and altered cell morphology

To decipher how dTTP metabolism and cell wall biosynthesis affect each other, we investigated the cellular function of RmlA, the cell wall biosynthetic enzyme that potentially links these two processes together using dTTP in the first step of the rhamnose biosynthetic pathway. As RmlA is essential in *Mycobacteria* [12], we could not knock out its gene. Alphey and his colleagues described a RmlA inhibitor [21] effective against *Pseudomonas aeruginosa* and *M. tuberculosis*. However, we could not detect any growth inhibition using this compound in *M. smegmatis*, even at the highest possible concentration limited by water solubility (200 μg/ml; 8-fold the MIC determined in *M. tuberculosis* [21]). Therefore, we chose to overexpress RmlA and investigate its effects in WT and PknG(-) *M. smegmatis* cells. RmlA activity has been described to be regulated by serine/threonine protein kinases PknB and PknG mediated phosphorylation. In PknG(-) cells, RmlA regulation by PknG is supposed to be switched off [17], i.e., RmlA is likely to be active in this strain. To overexpress the protein, we constructed a constitutive (RmlAc) and an inducible expression system (RmlAi). Protein expression in the latter case could be induced by adding TC to the growth medium. The degree of RmlA expression was verified by mRNA quantification using qPCR. sigA [31] and *Ffh* [27] were used as reference genes. We detected a 25-40-fold increase (p < 0.008) in the RmlA mRNA levels in the RmlAc and RmlAi strains upon TC induction, respectively (Fig 2A). We chose the inducible system for subsequent experiments. RmlAi *M. smegmatis* cultures grew at a WT rate (Fig 2B). Intriguingly, RmlA overexpression in the PknG (-) strain fully restored the growth arrest of the parental strain in a minimal medium (Fig 2B).

We investigated the possible phenotypes resulting from RmlA overexpression under the microscope. The distribution of cells on microscopy slides prepared from exponentially growing cultures indicates that under the same growth conditions (the same amount of detergent applied in the medium), RmlAi cells are more prone to form aggregates than WT cells (Fig 3). This observation is consistent with the noisiness of the RmlAi growth curves (Fig 2B).

The effects of RmlA overexpression on *M. smegmatis* morphology were investigated using phase contrast and epifluorescence microscopy (Fig 4). WT, PknG(-), and luciferase-expressing (Lxi) strains were used as controls to exclude artifacts potentially caused by TC and protein overexpression, respectively. We observed two distinct morphological changes upon the induction of RmlA overexpression in both WT and PknG(-) backgrounds. The cells became more variable in size and shape than the WT; on average, they became longer (Fig 4A). In addition, the normal rod shape of *M. smegmatis* was disrupted at the cell poles, especially at the 0.2 μg/ml TC concentration. Fig 4B shows that the cylindrical shape is changed into spherical at the tip of the cells. To statistically analyze the observations, we defined cells with a diameter larger than 0.6 μm deformed. In the RmlAi strain, 4.6% of the cells were deformed, 0

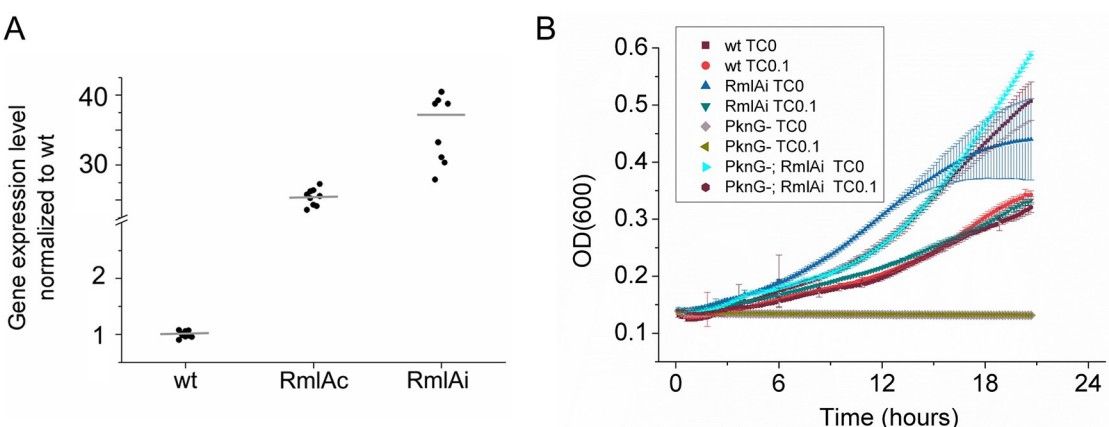

**Fig 2. RmlA overexpression in *M. smegmatis* and its effect on cell growth.** A) Overexpression was evaluated by measuring mRNA levels of RmlA in WT and overexpressing strains (RmlAi–induced overexpression by the addition of 0.05 µg /ml TC, or RmlAc—constitutive overexpression). RmlA levels in RmlAc and RmlAi strains were 25 and 40 fold higher (p < 0.008) than in the WT *M. smegmatis*, respectively. B) WT, RmlAi, PknG(-), and PknG(-); RmlAi *M. smegmatis* strains were grown in liquid culture without any treatment or with 0.1 µg/ml TC. OD (600) was measured every 10 min in a plate reader.

spherical (bubble-like) structure was observed (n = 119) at 0 µg/ml TC; 11% of the cells were deformed, spherical structures were observed in 9% of the cells (n = 86) at 0.1 µg/ml TC, and 52% of the cells were deformed, spherical structures were observed in 40% of the cells (n = 89) at 0.2 µg/ml TC. In the PknG(-); RmlAi strain, 3% of the cells were deformed, 0 spherical (bubble-like) structure was observed (n = 390) at 0 µg/ml TC; 11% of the cells were deformed, spherical structures were observed in 9% of the cells (n = 277) at 0.1 µg/ml TC, and 34% of the cells were deformed, spherical structures were observed in 34% of the cells (n = 259) at 0.2 µg/ml TC. In the control strains, 1–5% of the cells were deformed, with no spherical structure observed (n = 75–549 / strain).

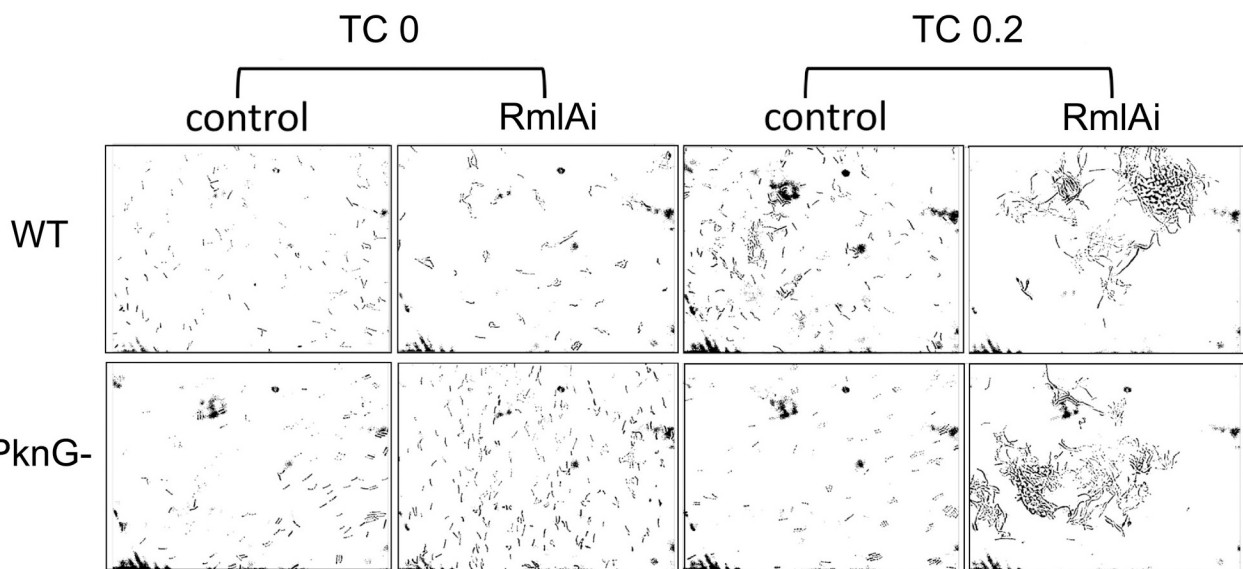

**Fig 3. RmlAi cells are more prone to aggregation.** The indicated strains were grown in liquid culture and streaked onto microscopy slides as described in Methods. RmlA overexpression was induced by the addition of 0.2 µg/ml TC. The control strains do not carry the RmlAi plasmid.

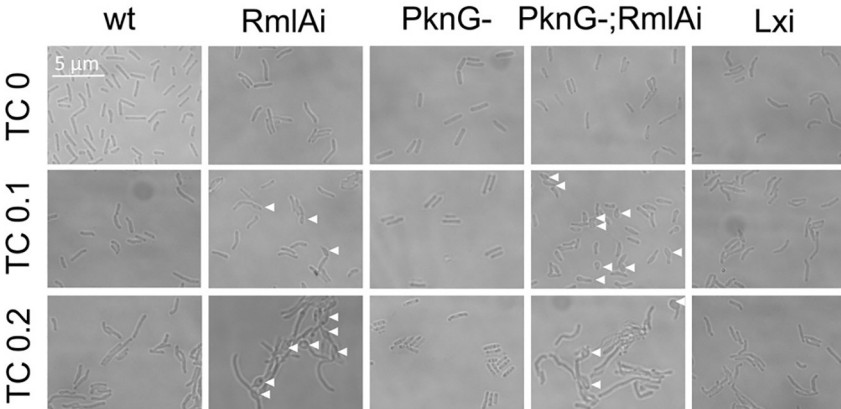

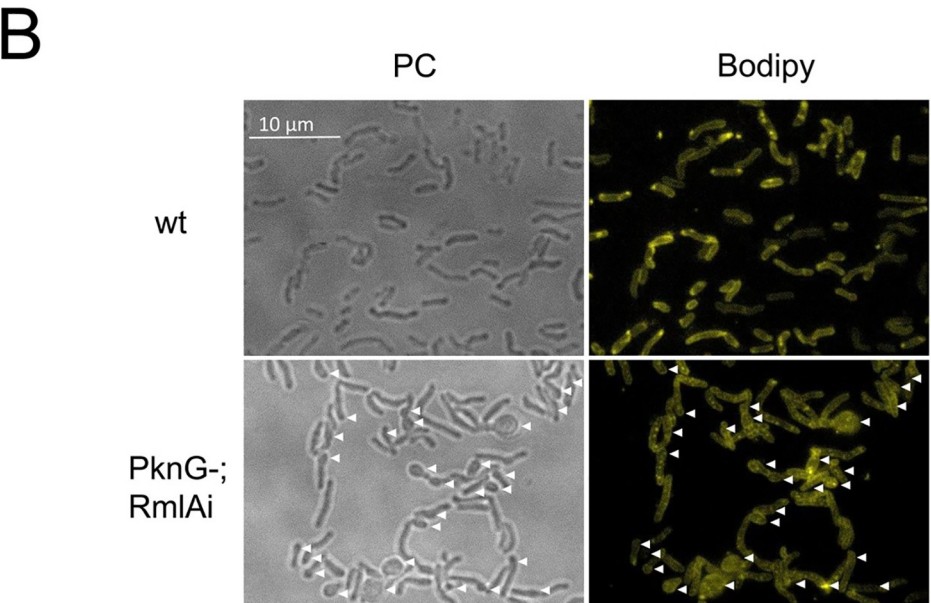

**Fig 4. Effects of RmlA overexpression on the morphology of *M. smegmatis*.** A) RmlA overexpression in both WT and PknG(-) backgrounds resulted in cell elongation and the appearance of spherical, bubble-like structures breaking off the normal rod shape. B) High-resolution visualization of the morphological changes in the RmlAi; PknG(-) strain upon TC induction. The cell membrane was stained by Bodipy 522/529. PC denotes phase-contrast images.

We quantified the observed elongation of the RmlAi cells and subjected the results to statistical analysis (Fig 5). We calculated 2.9 and 1.6 fold increase in the mean cell length in the RmlAi and RmlAi; PknG(-) strains compared to the WT, respectively.

## Effects of RmlA overexpression on the cellular dNTP pool

As RmlA uses dTTP to synthesize dTDP-rhamnose, we expected that the overexpression of RmlA could deplete dTTP from the cellular nucleotide pool, which, in turn, could cause

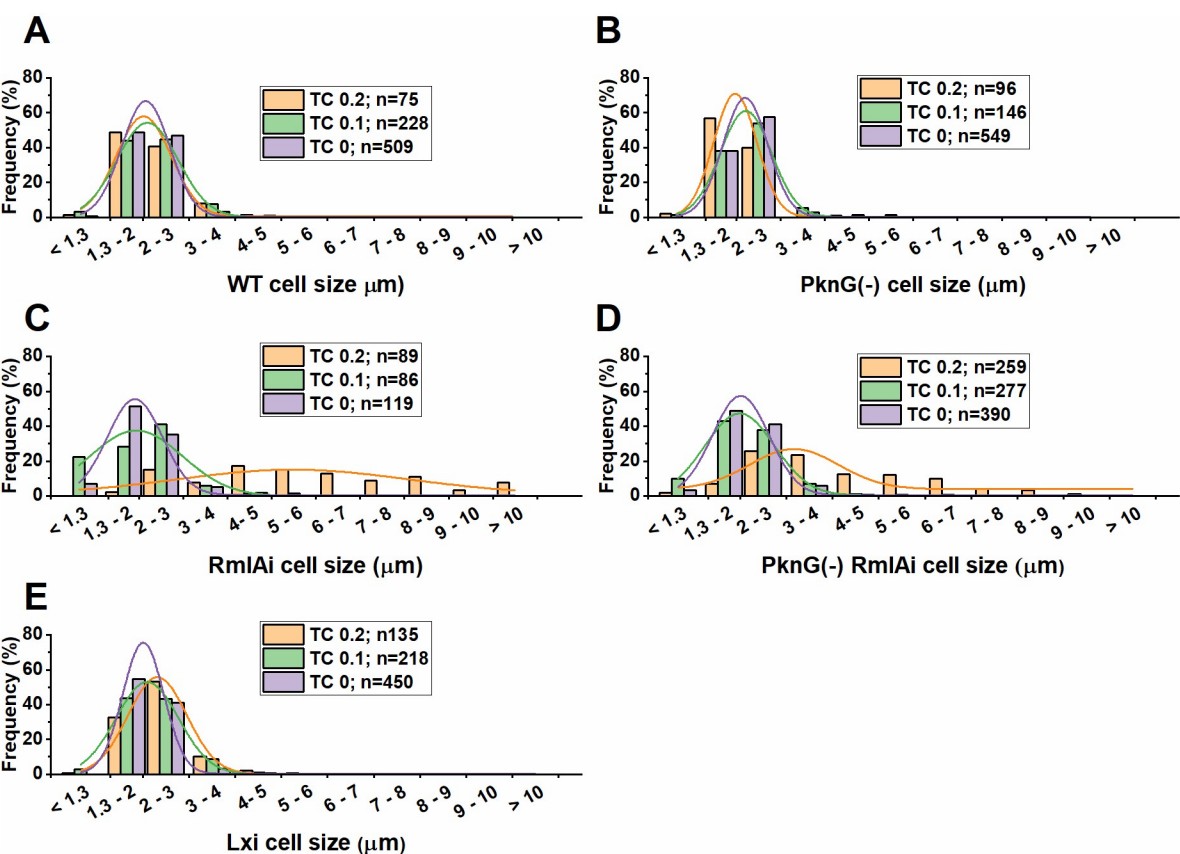

**Fig 5. Cell elongation upon RmlA overexpression.** The normalized distribution of cell length is shown as bars. Gaussian curve fitting to the data is shown as smooth lines. The number of cells counted in each sample (n) is shown in the legend. RmlA overexpression was induced by 0.1 or 0.2 µg/ml TC (TC 0.1 and TC 0.2 respectively). TC 0 stands for non-induced controls. The highest probability cell length yielded by the Gaussian curves in each case is the following in the order of 0.2 µg/ml TC; 0.1 µg/ml TC and no TC: 2.02±0.03 µm; 2.10±0.02 µm; 2.06 ±0.08 µm for the WT; 1.98±0.01 µm; 2.22±0.01 µm; 2.20±0.01 µm for the PknG(-); 5.48±0.53 µm; 1.94±0.16 µm; 1.91±0.02 µm for the RmlAi; 3.17±0.20 µm; 1.99±0.01 µm; 2.01±0.02 µm for the PknG(-); RmlAi and 2.32±0.01 µm; 2.10±0.02 µm; 2.01±0.02 µm for the Lxi samples.

replication defects. Therefore, we measured the cellular dNTP concentrations in all investigated strains using our improved fluorescence-based dNTP quantification assay (Fig 6) [30]. The principle of the assay is similar to that of the TaqMan assay. All four dNTPs are determined separately using dNTP-specific templates. The fluorescence output signal upon DNA elongation is directly proportional to the dNTP to be quantified. Interestingly, we could not detect any significant decrease in dTTP concentration caused by RmlA overexpression in the WT strain (Fig 6). However, RmlA overexpression in the PknG(-) background resulted in ~ 2 fold decrease in the dTTP concentration (p < 0.016) (Fig 6). Changes in the cellular dGTP, dATP, and dCTP concentrations were statistically insignificant (Fig 6).

## Effects of RmlA overexpression on replication and cell wall biosynthesis stress factors

RadA and LexA were chosen to indicate stress for DNA replication, while IniA and WhmD served as cell wall biosynthesis stress indicators. The upregulation of *iniA* and *whmD* upon treatment with the cell wall biosynthesis inhibitors EMB and INH [31, 32] was shown earlier and used as a positive control for cell wall biosynthesis stress. The upregulation of the SOS-

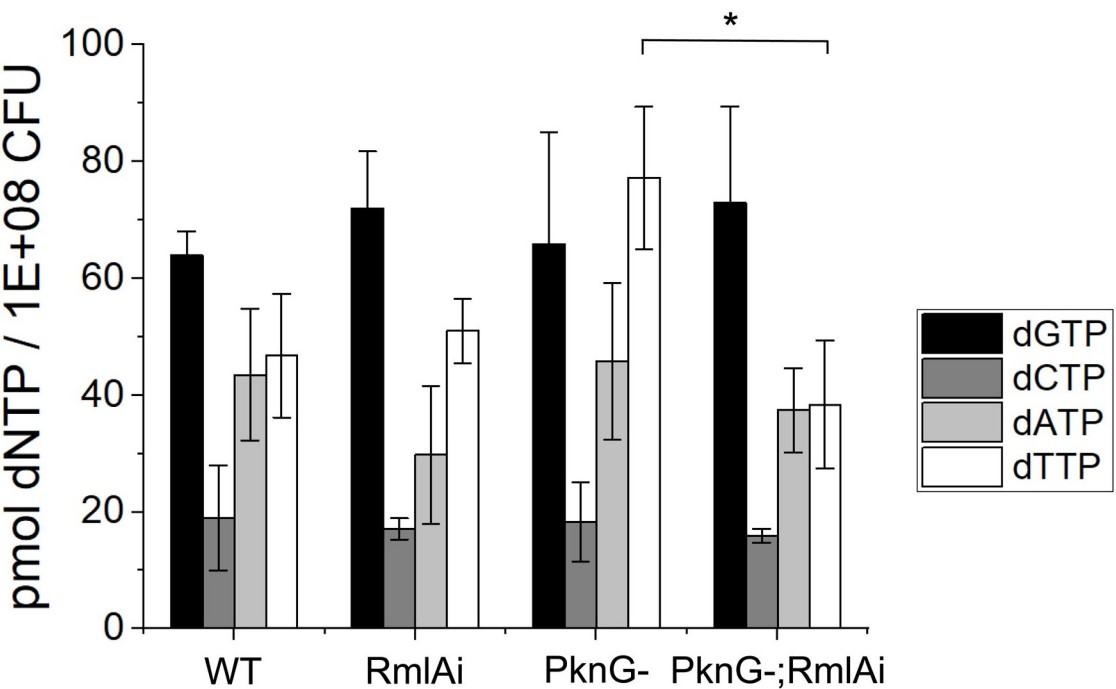

**Fig 6. The effect of RmlA overexpression on the cellular dNTP pool.** Concentrations of dNTPs in the extracts were measured according to [30]. The dTTP level in the RmlA overexpressing PknG(-) strain was significantly lower than that in the PknG(-) strain (p < 0.016), although it is unchanged when RmlA was overexpressed in the WT background. The cellular concentration of other dNTPs was not significantly changed in the RmlA overexpressing strains.

response protein LexA and the DNA repair protein RadA in response to CIP treatment was used as a positive control for replication stress [33, 34]. The concentrations of the applied drugs were chosen to inhibit cell growth while allowing downstream analysis of the remaining cells (Fig 7). To investigate the response of genes in the thymidylate biosynthesis pathway to RmlA overexpression, we measured the expression levels of Dcd:dut, ThyA, and ThyX enzymes. We also quantified the mRNA of the members of the rhamnose biosynthesis pathway (RmlA-D) and PknG known to regulate rhamnose biosynthesis [17]. The transcription level of these genes was measured by qPCR following different treatments or RmlA overexpression.

mRNA expression levels were normalized to that of the nontreated WT strain. RmlA overexpression had only a mild effect on the expression of the investigated genes in the WT background (Fig 8). However, in the PknG(-) background, all investigated stress factors were upregulated 2-6-fold (IniA 3.3-fold; WhmD 5.6-fold; LexA 2.15-fold and RadA 6-fold upregulation, respectively). We also detected a mild upregulation for RmlB and RmlC (~2-fold overexpression). However, RmlD was downregulated (-1.5-fold in the PknG-, and -2.5-fold in the WT background). The expression level of the thymidylate synthesis pathway enzymes was not changed significantly. The CIP treatment increased the expression of LexA (~13.5-fold), RadA (~12-fold), and IniA (~1.8-fold), while the EMB treatment resulted in RadA (~3.5-fold), WhmD (~5.5-fold), and IniA (~ 100-fold) overexpression. Interestingly, the expression level of RmlA was significantly decreased in the PknG(-) strain (-5.5-fold) and also as a result of the CIP treatment (-14-fold). In summary, the 2-6-fold increase in the expression of cell wall biosynthetic and replication stress factors indicates that RmlA overexpression induces quantifiable stress in the cellular processes studied.

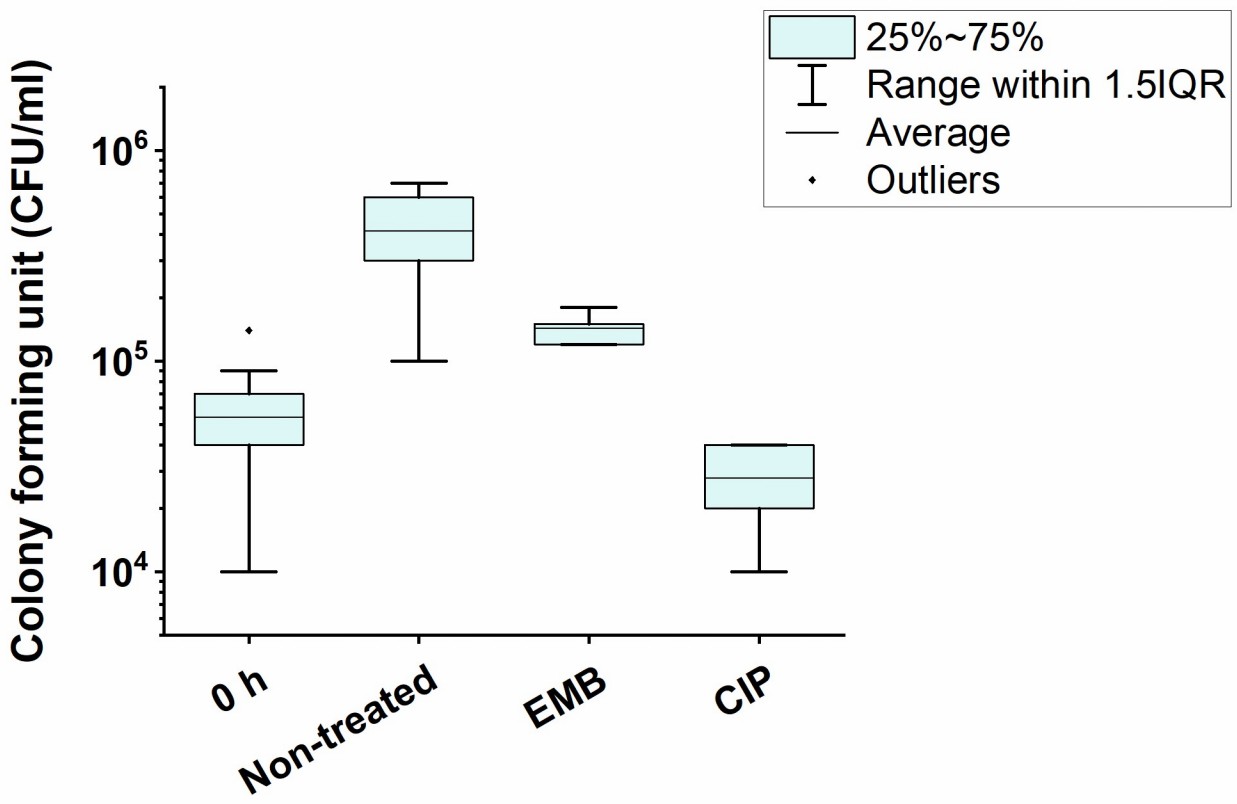

**Fig 7. Effects of CIP and EMB treatments on cell viability.** Treatments were performed on exponentially growing WT bacteria in liquid cultures for 8 h using 100 μg/ml EMB and 0.3 μg/ml CIP. CFU was counted on antibiotic-free agar plates. Concentrations were chosen so that growth inhibition was in the range of 20–80%.

### RmlA localizes nearby the cell membrane in a helix-like pattern

To investigate the cellular localization of RmlA, strains expressing the fluorescently tagged proteins RmlA-mOrange-2 or RmlA-GFP were constructed. To avoid artifacts from overexpression, the expression of these fluorescent constructs was driven by the RmlA promoter. We found that RmlA is accumulated at the cell perimeter in a helix-like pattern (Fig 9). The signal is most abundant at the tip of cells (Fig 9), where growth is thought to occur in mycobacteria.

### Discussion

To reveal possible interconnections between dTTP metabolism and cell wall biosynthesis, we set out to investigate the cellular function of RmlA. This enzyme uses dTTP as a precursor in the rhamnose biosynthetic pathway. Qu and his colleagues showed [12] that RmlA is essential in mycobacteria and constructed a *M. smegmatis rmlA* null strain that conditionally expressed RmlA from *M. tuberculosis*. At a permissive temperature, cells exhibited the normal rod shape. In contrast, cells showed irregular surface wrinkles, and subsequent lysis at a non-permissive temperature at which the plasmid expressing *M. tuberculosis* RmlA could not replicate [12]. The morphological changes we observed upon RmlA overexpression are in line with those resulting from *rmlA* depletion [12]. Our RmlA overexpressing cells also appeared significantly longer than the WT, indicating a role of RmlA in cell growth. Furthermore, 40% of the RmlA overexpressing cells developed spheroidal structures at the cell poles. These results indicate

| Target | Sample | RmlAi | PknG- | PknG-;RmlAi | controls EMB | controls CIP |
|---|---|---|---|---|---|---|
| rhamnose biosynth. | rmlA | 38.23* | -5.50 | 39.94 | -1.09 | -14.20 |
| | rmlB | -1.30 | 1.05 | 2.18 | -1.31 | -2.03 |
| | rmlC | -1.60 | -1.56 | 1.92 | -1.64 | -2.28 |
| | rmlD | -2.43 | -3.51 | -1.54 | -1.01 | -2.88 |
| regulation | pknG | 1.44 | <D.L.* | <D.L.* | -2.03 | -2.22 |
| stress factors | iniA | -1.06 | -1.17 | 3.34 * | 100.19* | 1.83 |
| | whmD | 1.39 | 1.97 * | 4.58 * | 5.53 * | -1.99 |
| dTTP biosynth. | dcddut | -2.11 | -4.18 | 1.78 | -1.02 | -1.30 |
| | thyA | -1.14 | -1.88 | -1.33 | -5.32 | -6.51 |
| | thyX | -2.49 | -2.43 | 1.05 | -3.00 | -4.35 |
| stress factors | lexA | 1.12 | 1.14 | 2.15 * | -1.10 | 13.48* |
| | radA | -1.22 | 2.50 * | 5.96 | 3.57 | 12.23* |

**Fig 8. Effects of RmlA overexpression on the expression of different genes.** RadA and LexA were used as indicators for replication stress, while IniA and WhmD indicated cell wall biosynthesis stress. Dcd:dut, ThyA, and ThyX represent the thymidylate biosynthesis pathway, while RmlA-D belongs to the rhamnose biosynthetic pathway. PknG, known to regulate rhamnose biosynthesis, was also measured. The following treatments were used as positive controls: CIP for replication stress and EMB for cell wall biosynthesis stress. Expression levels were normalized to those of the WT strain. Changes in expression levels are shown as a heat map. Blue depicts downregulation, while red is for upregulation. Black star indicates $p < 0.05$ for the change in expression.

that both the absence and the excess of RmlA activity cause morphological abnormalities in *M. smegmatis*.

We found that RmlA localizes in a helical pattern at the cylindrical part of the cell in addition to its enrichment at the poles. This pattern indicates that crosslinking of peptidoglycan and arabinogalactan layers is not restricted to the cell poles despite the observation that newly synthesized peptidoglycan in mycobacteria is limited to the poles [35]. The helical localization pattern of RmlA is reminiscent of the helical cables formed by cytoskeletal proteins (FtsZ, MreB, and Mbl) in rod-shaped bacteria that grow via cylindrical extension [36–38]. In the absence of MreB homologs in mycobacteria [39], the rod shape must be achieved by yet unknown mechanisms. The observed disrupted rod morphology at the poles in the RmlAi strains and the localization pattern of RmlA-GFP using epifluorescence microscopy are also reminiscent of what Plocinski et al. detected when studying the cell wall synthesis protein *cwsA* at a similar resolution [40, 41].

Since RmlA uses dTTP to synthesize dTDP-D-glucose, it could potentially deplete the cellular dTTP pool causing replication defects and growth inhibition if not appropriately regulated, as shown with other dTTP deficient conditions [42]. However, we could not detect any significant growth arrest in RmlA overexpressing strains. On the contrary, the RmlA-overexpressing PknG(-) strain grew better than its parental strain in a minimal medium. The reduced growth of the PknG-deficient *M. tuberculosis* strain in the minimal medium was shown in an earlier report [43], in which Cowley et al. proposed that growth arrest is caused by a decrease in the *de novo* glutamine synthesis [43]. Interestingly, in our experiments, RmlA overexpression fully rescued the growth arrest observed in PknG(-) *M. smegmatis* in a minimal medium. We also

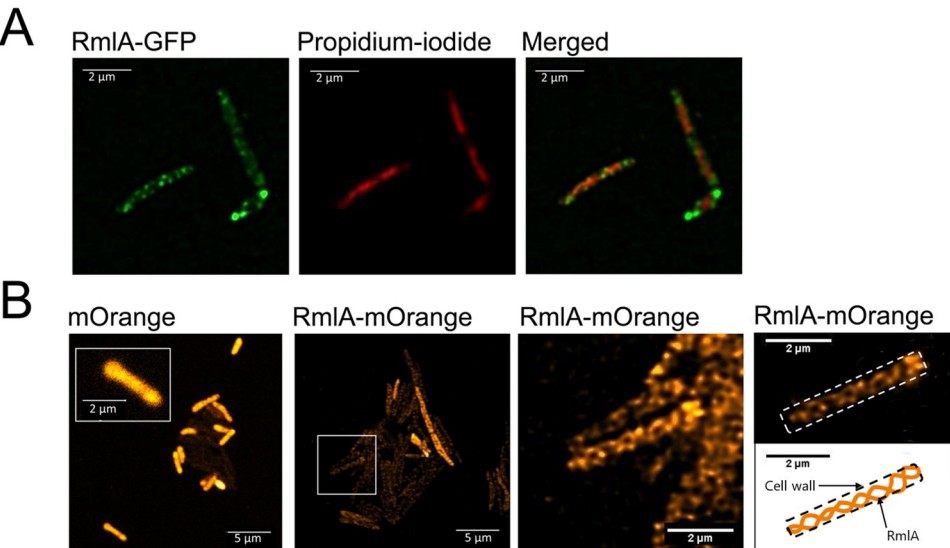

**Fig 9. The cellular localization pattern of fluorescently tagged RmlA.** To investigate the cellular localization of RmlA, we constructed *M. smegmatis* strains expressing GFP (A) and mOrange-2 (B) tagged RmlA under the control of the endogenous RmlA promoter. A) The localization pattern of the GFP tagged protein was observed using confocal microscopy. Of the 57 cells analyzed, all exhibited a helical fluorescence pattern here. DNA was stained with propidium iodide. B) The localization of mOrange-2 tagged protein was investigated using STED super-resolution imaging. We found that in all 68 cells analyzed, RmlA localized to the cell perimeter in a helix-like pattern, while no helicity was observed in the mOrange controls (mOrange panel in Fig 9B). Cell boundaries are indicated with a dashed line.

found that the RmlA overexpressing strains (both RmlAi and PknG(-); RmlAi) are more prone to aggregation than parental strains. Hsu et al. and Hardy et al. showed that RmlA influences biofilm formation in *Listeria monocytogenes* [10] and *Caulobacter crescentus* [11], respectively. A similar influence of RmlA in *M. smegmatis* would be consistent with the observed increase in aggregation. The importance of RmlA regulation by phosphorylation is also discernible from the results on cellular dTTP concentrations in the RmlA overexpressing strains. While we could not detect any significant cellular dTTP concentration change in PknG proficient cells, the dTTP level decreased to its half upon RmlA overexpression in the PknG(-) background, devoid of negative regulation by PknG phosphorylation. Similarly, RmlA overexpression only resulted in significant expression changes in the PknG(-) background. In addition to its PknG regulation, the feedback inhibition of RmlA also seems to be efficient in avoiding dTTP depletion. In a 40-fold excess of RmlA in the RmlAi;PknG(-) cells, dTTP should have been entirely depleted without feedback inhibition. Moretti et al. showed that *Salmonella enterica typhimurium* RmlA could use all dNTPs and NTPs *in vitro* [44], although with a dTTP bias. We found that *M. smegmatis* RmlA perturbed the cellular concentration of only dTTP without affecting other dNTPs, which suggests that it is specific for dTTP in the cellular environment.

RmlA overexpression resulted in both cell wall biosynthetic and replication stress indicated by the 2-6-fold moderate but solid increase in the mRNA levels of the selected stress factors (Fig 8). Although severe dNTP depletion was not observed in cell extracts, transient dTTP depletion in the cells may have led to thymine starvation which can induce replication stress.

Due to the efficient feedback inhibition mechanism of RmlA, we could not establish at what point dTTP availability may serve as a link between replication and cell wall biosynthesis. We observed, however, that both cell wall biosynthesis and replication stresses can be moderately

induced by RmlA overexpression, and there might be some interconnection between the two processes. On the other hand, our investigations unveiled that the cylindrical part of the cell may not be as inert as previously thought. The fact that RmlA, acting to crosslink the peptidoglycan and arabinogalactan layers, localizes throughout the whole cell length in a helical pattern strongly suggests that cell wall synthesis also occurs in the cylindrical part of the cell, not only at the poles. Consistent with this, the morphological changes upon RmlA overexpression indicate that RmlA plays a role in determining cell shape driven by a yet unknown mechanism in mycobacteria.

## Supporting information

**S1 File. The "raw data" archive contains the sequences of oligos used in this study, raw data for qPCR, dNTP quantitation, growth measurements, microscopy images, and data for the determination of cell length distribution.**
(ZIP)

**S2 File.**
(DOCX)

## Acknowledgments

We thank Krisztina Németh for fluorescence imaging trials.

## Author Contributions

**Conceptualization:** Rita Hirmondó, Judit Tóth.

**Formal analysis:** Rita Hirmondó, Dániel Molnár.

**Funding acquisition:** Judit Tóth.

**Investigation:** Rita Hirmondó, Ármin Horváth, Dániel Molnár, György Török.

**Resources:** Liem Nguyen.

**Supervision:** Rita Hirmondó, Judit Tóth.

**Writing – original draft:** Rita Hirmondó, Judit Tóth.

**Writing – review & editing:** Rita Hirmondó, Ármin Horváth, Dániel Molnár, György Török, Liem Nguyen, Judit Tóth.

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
