## [Decision Letter · Decision Letter 0]

13 Oct 2021

PONE-D-21-26930Perturbation of rhamnose biosynthesis reveals crosstalk between mycobacterial cell wall synthesis and DNA replicationPLOS ONE

Dear Dr. Tóth,

Thank you for submitting your manuscript to PLOS ONE. After careful consideration, we feel that it has merit but does not fully meet PLOS ONE’s publication criteria as it currently stands. Therefore, we invite you to submit a thoroughly revised version of the manuscript that addresses the points raised during the review process (see below).

We look forward to receiving your revised manuscript.

Kind regards,

Olivier Neyrolles

Academic Editor

PLOS ONE

Journal Requirements:

Reviewers' comments:

Reviewer's Responses to Questions

**Comments to the Author**

1. Is the manuscript technically sound, and do the data support the conclusions?

Reviewer #1: Partly

Reviewer #2: Partly

2. Has the statistical analysis been performed appropriately and rigorously? 

Reviewer #1: Yes

Reviewer #2: No

3. Have the authors made all data underlying the findings in their manuscript fully available?

Reviewer #1: Yes

Reviewer #2: Yes

4. Is the manuscript presented in an intelligible fashion and written in standard English?

Reviewer #1: Yes

Reviewer #2: No

5. Review Comments to the Author

Reviewer #1: This paper describes the construction and characterization of an overexpression strain of RmlA in M. smegmatis. Overall, the experiments were conducted well with appropriate controls. However the conclusions are not supported by the data. The manuscript should either be strengthened with additional data or the overstated claims removed.

Major Points

L43. Essentiality of a gene is not a “strong basis” for the development of anti-tubercular agents. This is a gross over-statement.

L62 There is no evidence that RmlA is a checkpoint, this is a stretch from the published data. It is a hypothesis.

L307 No measurements of cell length were presented to support the statement that they were significantly longer. the link to cell growth division is weak.

The authors refer to extensive clumping in their cultures. What was the rationale for culturing M. smegmatis without Tween 80, as is the norm in the field to reduce aggregation? If clumping was an issue, how was OD determined reliably?

The changes in gene expression for the stress response are marginal. Although they achieved statistical significance, the magnitude of the change was very small and much lower than the response to the control agents. Therefore, the claim that RmlA overexpression induces these stresses is not supported by the data. The gene expression studies did not include any no RT controls to account for the presence of genomic DNA.

For the experiments with CIP and EMB, how was the concentration used determined/selected?

Some more explanation of the methods to measure dNTPs is needed. It was not clear. In the results how this was done.

The discussion was very lengthy. It should be restricted to a discussion of the results in context of the published literature instead of a literature review. For example, Lines 311 to 361 read like a literature review.

L388 The conclusion that RmlA overexpression leads to cell wall and replication stress is not supported by the data in the paper. In fact, it's more likely that RmlA overexpression does not lead to cell wall or replication stress based on the very small changes in gene expression. the discussion should be revised to remove the overstated claims and limit discussion to what is shown in the data.

The supporting information should be included in the main body of the paper as it is required to follow the methods.

Minor points

There was a lot of unscientific language and claims which were not supported by refernces e.g.

L23 what is “outstanding genetic adaptability”?

L24 “most dreaded member”

L144 what does “2-2 stress”mean?

References are missing to support the statements in L24, L38.

L193 what does the “X’ stand for? What was the maximum concentration tested against M. smegmatis?

L206 “somewhat disturbed by the intensive clumping”

L240 what does “quasi unchanged” mean?

L198 and others “constitutive” not “constant”

L245 Reference in wrong format

L330 is “bulgy” a real word?

L87-90 The methods section should describe the methods, not the rationale or interpretation of data. Why did the authors use TC and not anhydrotetracycline?

L101. Table legend should just be the legend.

Abbreviations are inconsistent. For example, why is M. smegmatis used and not M. tuberculosis? Why is Cip lower case, but EMB upper case. Why is there an abbreviation for EMB but no isoniazid?

Abbreviations should be defined at first use.

Figure 1 is not required as the data do not support this conclusion. It was also not written in English.

Reviewer #2: In this manuscript entitled “Perturbation of rhamnose biosynthesis reveals crosstalk between mycobacterial cell wall synthesis and DNA replication” Hirmondo et al. have attempted to elucidate the crosstalk between cell wall synthesis and DNA replication in M. smegmatis, by overexpressing RmlA. RmlA, is an essential gene synthesizing L-Rhamnosyl residues using D-glucose-1-phosphate (G1P) and dTTP as the precursors. The rhamnosyl residues synthesised are critical to the structural integrity of the mycobacterial cell wall. They have investigated the role of RmlA in the biosynthesis and assembly of cell wall in M.smegmatis by RmlA overexpression which caused morphological changes including cell elongation and the appearance of polar bulbs at the tip of the cell. Although the over expression of RmlA in PknG deficientM. smegmatis results in significant dTTP depletion, intracellular dTTP concentration was not affected which lead them to suggest a role for PknG in the regulation of dTTP uptake. However, the manuscript and the experiments presented(including grammatical and spelling errors) do not adequately justify the title and the abstract.

1. In the conditional knockdown of RmlA (Qu et al) the effect on cellular morphology has been investigated. The overexpression levels of rmlA transcript (40 fold in constitutive and 80 fold in inducible expression) is way too high to see physiological effect. At protein level (not estimated) it could be much more. The overexpression levels should be between 5 to 10 fold to be physiologically relevant. Either they could use the knock down strain of Qu et al or generate a conditional knockdown of rmlA using a CRISPR-Cas or another approach to address how rmlA affects cell wall bio-synthesis pathway and cellular dNTP pool.

2.What is the effect of over expression of RmlA on the expression levels of Rml B-D? The connection between replication and cell wall biosynthesis needs to be clearly demonstrated. Increase in transcript level of replication stress factors like whmD and iniA upon RmlA overexpression does not establish the connection between cell wall synthesis and replication. Increase in lexA would imply SOS connection?

3. Increase in RmlA would mean higher utilization of dTTP. Although they claim that there is no depletion of dTTP, transient depletion may lead to thymine starvation which can induce stress.

4. In Figure 3, the cells depicting bulging can should be quantitated and a zoomed clear figure of the bulged cells would be appropriate. The phase contrast images are not in focus and should be improved. Similarly in both confocal and super resolution microscopy (Fig. 6), the number of cells analyzed should be provided with statistical analyses. Two cells in RmlA m-orange panel are looking like cells in m-orange panel, with no helical pattern. Statistical data for helicaly patterned RmlA containing cells should be provided. How does localization of RmlA affects the conclusion of the authors linking cell wall biosynthesis and replication is not clear.

5. Line 204 – 208 : Tween 80 concentration can be increased (upto 0.5%) to ensure clumping is reduced, to obtain better growth patterns. Line 206 -208: vaguely written.

I am listing a few more of corrections. There are more.

1. Line 71: ‘that’ should be ‘this’.

2. Line 74: ‘overexpression’ is a better word

3. Line 80 should be: an M.smegmatis PknG knock out strain.

4. In Materials and methods: write the methods only and not why a method was used. You can provide the rationale for the experiment in Results section.

5. Line 140-141: Rephrase

6. Line 144: rephrase ‘2-2 stress factors’

7. Line 153,154: ‘Quigene’ should be ‘Qiagen’.

8. Lines 207-209: vaguely written; Rephrase

9. Line 213: constant should be ‘constitutive’.

10. Fig 4 legend: No need to explain in the method in figure legend.

6. PLOS authors have the option to publish the peer review history of their article (what does this mean?). If published, this will include your full peer review and any attached files.

Reviewer #1: No

Reviewer #2: No

---

## [Author Response · Author response to Decision Letter 0]

7 Jan 2022

Comments to the Author

Reviewer #1: This paper describes the construction and characterization of an overexpression strain of RmlA in M. smegmatis. Overall, the experiments were conducted well with appropriate controls. However the conclusions are not supported by the data. The manuscript should either be strengthened with additional data or the overstated claims removed.

Thank you for your constructive comments. In the revised version, we provided additional data and removed claims that are not fully supported by the data.

Major Points

L43. Essentiality of a gene is not a “strong basis” for the development of anti-tubercular agents. This is a gross over-statement.

We agree that essentiality alone is not a “strong basis” for drug development. We originally wrote that the essentiality of the gene together with the absence of a similar gene product in the host provides a strong basis. 

Following your comment, we rephrased the message to “potentially valid drug target” in order to avoid any overstatement.

We also searched for the definition of a good drug target. According to a recent paper in Nat Rev Drug Discovery:

“Relevant characteristics of microbial drug targets include: essentiality for microbial growth and survival; pharmacological tractability and accessibility; similarity to related mammalian molecules; presence in important pathogens; potential for the development of resistance; and lack of target-based cross-resistance.” (Emmerich et al.: Improving target assessment in biomedical research: the GOT-IT recommendations. 20, 64–81 (2021). https://doi.org/10.1038/s41573-020-0087-3)

Considering these relevant characteristics, RmlA may qualify as a valid target. It was mentioned so recently here, as well: Qu D, Zhao X, Sun Y, Wu F-L and Tao S-C (2021) Mycobacterium tuberculosis Thymidylyltransferase RmlA Is Negatively Regulated by Ser/Thr Protein Kinase PknB. Front. Microbiol. 12:643951. doi: 10.3389/fmicb.2021.643951

L62 There is no evidence that RmlA is a checkpoint, this is a stretch from the published data. It is a hypothesis.

We corrected this and stated that it is a hypothesis. 

L307 No measurements of cell length were presented to support the statement that they were significantly longer. the link to cell growth division is weak.

We quantified the length of ~ 3700 cells and now provide thorough statistics for cell length distribution in the various strains in Fig 5.

Reference to division regulation is removed.

The authors refer to extensive clumping in their cultures. What was the rationale for culturing M. smegmatis without Tween 80, as is the norm in the field to reduce aggregation? If clumping was an issue, how was OD determined reliably?

We did cultivate smegmatis cells in Tween 80 as described in Materials and methods. Extensive clumping happened only upon RmlA overexpression in spite of the presence of Tween in the medium. We added this comment to the text. 

Although the OD measurements were noisier than in the WT cell cultures, the aggregation did not compromise the experiments. We consider the increased tendency to aggregate a phenotype upon RmlA overexpression, consistent with the fact that cell wall biosynthesis is perturbed.

The changes in gene expression for the stress response are marginal. Although they achieved statistical significance, the magnitude of the change was very small and much lower than the response to the control agents. Therefore, the claim that RmlA overexpression induces these stresses is not supported by the data.

We agree that changes upon RmlA overexpression in the WT strain are small. As we observed dTTP decrease only in the PknG-;RmlAi strain and not in the WT strain, we expected that the effect of RmlA overexpression in the PknG-;RmlAi strain could be larger. We, therefore, repeated the measurement and obtained, indeed, a larger change in the expression levels of all investigated stress factors (Fig 8). In the case of the established stress response genes whmD and radA, the measured overexpression in the PknG(-)/RmlAi strain is commensurable with that of the positive controls. The M. tuberculosis homologue of WhmD, Whib2 is overexpressed 2-3-fold in response to EMB and INH as shown below (Geiman et al., (2006) Differential Gene Expression in Response to Exposure to Antimycobacterial Agents and Other Stress Conditions among Seven Mycobacterium tuberculosis whiB-Like Genes, ANTIMICROBIAL AGENTS AND CHEMOTHERAPY, Aug. 2006, p. 2836–2841, doi:10.1128/AAC.00295-06):

The 3.3-fold overexpression of iniA in the PknG(-)/RmlAi strain is commensurable with the effect of induction by ampicillin, and by the combination B-lactam–B-lactamase inhibitor Unasyn (ampicillin/sulbactam), which was consistently three- to fivefold greater than in control cultures (Alland et al., (2000) Characterization of the Mycobacterium tuberculosis iniBAC Promoter, a Promoter That Responds to Cell Wall Biosynthesis Inhibition, JOURNAL OF BACTERIOLOGY, doi: 10.1128/JB.182.7.1802-1811.2000.). Following gamma irradiation (!), 2.5-fold increase in the mRNA level of radA was found in Pyrococcus furiosus (Williams et al., (2007) Microarray analysis of the hyperthermophilic archaeon Pyrococcus furiosus exposed to gamma irradiation, Extremophiles, doi: 10.1007/s00792-006-0002-9.) to be compared to our 6-fold increase in the PknG(-)/RmlAi strain. In view of these data, we hope that the Reviewer agrees that the measured stress response in our experiments is relevant in addition to being statistically significant.

The gene expression studies did not include any no RT controls to account for the presence of genomic DNA. 

We now included the no RT and no template controls in the excel table submitted in the supplementary zip file as part of the data availability requirements.

For the experiments with CIP and EMB, how was the concentration used determined/selected?

We scanned a range of drug concentrations with the objective to arrive at a level of growth inhibition between 20-80 % in order to have enough cells left to analyse. We included Fig 7 in the manuscript showing that the selected concentrations were in this regime. 

Some more explanation of the methods to measure dNTPs is needed. It was not clear. In the results how this was done.

We now included a brief description of the principle of the method used in addition to the reference for the published method: 

“The principle of the assay is similar to that of the TaqMan assay. All four dNTPs are determined separately using dNTP-specific templates. The fluorescence output signal upon DNA elongation is directly proportional to the dNTP to be quantified.”

The discussion was very lengthy. It should be restricted to a discussion of the results in context of the published literature instead of a literature review. For example, Lines 311 to 361 read like a literature review.

We agree and now restrict the discussion to subjects immediately pertinent to the results.

L388 The conclusion that RmlA overexpression leads to cell wall and replication stress is not supported by the data in the paper. In fact, it's more likely that RmlA overexpression does not lead to cell wall or replication stress based on the very small changes in gene expression. the discussion should be revised to remove the overstated claims and limit discussion to what is shown in the data.

As RmlA overexpression in the WT background did not result in solid replication and cell wall stress, we repeated the measurement in the PknG-;RmlAi strain, as well. In the case of the established stress response genes whmD and radA, the measured overexpression in the PknG(-)/RmlAi strain is commensurable with that of the positive controls (Fig 8). Please see above a longer version of this response with references.

Nevertheless, we revised the text and removed overstated claims. 

The supporting information should be included in the main body of the paper as it is required to follow the methods.

Following your suggestion, we included the supporting figures in the main body of the paper (Fig 2B and Fig 3 in the revised version).

Minor points. There was a lot of unscientific language and claims which were not supported by refernces e.g. 

Thank you for pointing these out. We reworked the text extensively.

L23 what is “outstanding genetic adaptability”? 

This expression was not necessary to support the claim and was removed.

L24 “most dreaded member”

Removed.

L144 what does “2-2 stress”mean? 

The “2-2 stress factors” referred to RadA and LexA, and IniA and WhmD. The sentence is rephrased for clarity.

References are missing to support the statements in L24, L38.

References are now added.

L193 what does the “X’ stand for? What was the maximum concentration tested against M. smegmatis?

X remained in the manuscript by error. We used the maximum concentration limited by water solubility: 200 µg/ml. This is 8-fold the MIC determined in M. tuberculosis.

L206 “somewhat disturbed by the intensive clumping”

Expression replaced using scientific language. 

L240 what does “quasi unchanged” mean?

Expression replaced using clear statements.

L198 and others “constitutive” not “constant”

Corrected. 

L245 Reference in wrong format

Reference corrected. 

L330 is “bulgy” a real word?

Replaced using scientific language. 

L87-90 The methods section should describe the methods, not the rationale or interpretation of data. 

We removed rationales and interpretations. 

Why did the authors use TC and not anhydrotetracycline?

In Williams et al: Improved Mycobacterial Tetracycline Inducible Vectors (https://www.ncbi.nlm.nih.gov/pmc/articles/PMC3495547/#!po=73.5294) (Fig 3), the induction of protein expression was investigated using TC, ATC, and doxycycline. TC proved to be the most efficient of the three, and therefore, we also used TC.

L101. Table legend should just be the legend.

Corrected.

Abbreviations are inconsistent. For example, why is M. smegmatis used and not M. tuberculosis? Why is Cip lower case, but EMB upper case. Why is there an abbreviation for EMB but no isoniazid?

Corrected.

Abbreviations should be defined at first use.

Corrected.

Figure 1 is not required as the data do not support this conclusion.

This figure presents in a transparent way the hypothesis we examined. It does not contain any suggestions to the conclusions. We highlighted in the figure legend that this is a hypothesis.

Reviewer #2: In this manuscript entitled “Perturbation of rhamnose biosynthesis reveals crosstalk between mycobacterial cell wall synthesis and DNA replication” Hirmondo et al. have attempted to elucidate the crosstalk between cell wall synthesis and DNA replication in M. smegmatis, by overexpressing RmlA. RmlA, is an essential gene synthesizing L-Rhamnosyl residues using D-glucose-1-phosphate (G1P) and dTTP as the precursors. The rhamnosyl residues synthesised are critical to the structural integrity of the mycobacterial cell wall. They have investigated the role of RmlA in the biosynthesis and assembly of cell wall in M.smegmatis by RmlA overexpression which caused morphological changes including cell elongation and the appearance of polar bulbs at the tip of the cell. Although the over expression of RmlA in PknG deficientM. smegmatis results in significant dTTP depletion, intracellular dTTP concentration was not affected which lead them to suggest a role for PknG in the regulation of dTTP uptake. However, the manuscript and the experiments presented(including grammatical and spelling errors) do not adequately justify the title and the abstract.

Thank you for your insightful comments. In the revised version, we removed overstated claims and changed the title and abstract accordingly. 

1. In the conditional knockdown of RmlA (Qu et al) the effect on cellular morphology has been investigated. The overexpression levels of rmlA transcript (40 fold in constitutive and 80 fold in inducible expression) is way too high to see physiological effect. At protein level (not estimated) it could be much more. The overexpression levels should be between 5 to 10 fold to be physiologically relevant. 

It is not clear to us what the Reviewer means by physiological in this case. When we perturb a system to investigate the role of a protein, we intentionally move away from physiologically relevant cellular concentrations. Using KO strains represents the extreme of this logic and is yet entirely accepted as a mode of perturbation.

Either they could use the knock down strain of Qu et al

This was our original intention. We contacted Prof Ma and his colleagues several times with no response (see the request for the strain in the screenshot below). That is when we changed our strategy.

or generate a conditional knockdown of rmlA using a CRISPR-Cas or another approach to address how rmlA affects cell wall bio-synthesis pathway and cellular dNTP pool.

This is a very good approach. The reason why we chose another one after several trials to ask for the knock-down already generated by Qu et al is that we were primarily interested in the dNTP pool and in the potentially resulting replication stress. Therefore, we thought that overexpression and in consequence, the expected depletion of dTTP could be a good tool to investigate the formulated hypothesis. Looking at the morphological changes resulting from RmlA overexpression, it still seems like a valid tool for the perturbation of the relevant physiological pathways. 

2.What is the effect of over expression of RmlA on the expression levels of Rml B-D?

We completed the experiment with the inclusion of the RmlB-D transcripts, as suggested. RmlA overexpression did not significantly change the level of RmlB-D mRNAs (Fig 8). 

In the repeated experiment, we changed the 16S RNA reference gene to Ffh as we found the latter to be more stable in this experimental system. The change of one of the reference genes naturally modified fold-change values but not tendencies and conclusions.

The connection between replication and cell wall biosynthesis needs to be clearly demonstrated. Increase in transcript level of replication stress factors like whmD and iniA upon RmlA overexpression does not establish the connection between cell wall synthesis and replication.

In the revised version, we removed overstatements. 

 Increase in lexA would imply SOS connection?

Yes. Replication stress generally induces SOS response.

3. Increase in RmlA would mean higher utilization of dTTP. Although they claim that there is no depletion of dTTP, transient depletion may lead to thymine starvation which can induce stress.

Thank you for this comment; we included it in the text.

4. In Figure 3, the cells depicting bulging can should be quantitated and a zoomed clear figure of the bulged cells would be appropriate.

We conducted further microscopy experiments to better show this change in cell morphology (currently Fig 4). We also included fluorescence imaging to show morphology (Fig 4B) and quantitated all our cell biology experiments. 

The phase contrast images are not in focus and should be improved.

We tried our best to achieve high-quality images using a microscopy objective HCX PL FLUOTAR 100x; NA.=1.30 OIL. The average length of M. smegmatis in our cultures is 2 microns, the average width is 0.5 microns. The width of the cells is therefore only 2-3x larger than the resolution limit of our system. In our view, this technical limitation does not hinder the clear presentation of the phenotype upon RmlA overexpression.

Similarly in both confocal and super resolution microscopy (Fig. 6), the number of cells analyzed should be provided with statistical analyses.

We agree and now provide the requested statistical analyses. 

 Two cells in RmlA m-orange panel are looking like cells in m-orange panel, with no helical pattern. 

The referred cells are overexposed in the settings presented. Setting a lower gain to visualize these cells, we obtain the same helical pattern as in the other cells chosen for demonstration. Both the low and high gain pictures are available in the row data zip file. The presented statistical analysis now shows that the helical pattern is present in 100% of the cells visualized.

Statistical data for helicaly patterned RmlA containing cells should be provided.

Data are now provided. 

How does localization of RmlA affects the conclusion of the authors linking cell wall biosynthesis and replication is not clear.

The currently available data do not establish any linkage between the subcellular localization of RmlA and its potential role in linking cell wall biosynthesis and replication. We rephrased the text so that our position on this issue is clear.

5. Line 204 – 208 : Tween 80 concentration can be increased (upto 0.5%) to ensure clumping is reduced, to obtain better growthpatterns. 

Although the OD measurements were noisier than in the WT cell cultures, the aggregation did not compromise the experiments. We consider the increased tendency to aggregate a phenotype upon RmlA overexpression consistent with the fact that cell wall biosynthesis is perturbed.

Line 206 -208: vaguely written.

Replaced using scientific language.

I am listing a few more of corrections. There are more.

1. Line 71: ‘that’ should be ‘this’.

Corrected.

2. Line 74: ‘overexpression’ is a better word

Corrected.

3. Line 80 should be: an M.smegmatis PknG knock out strain.

Corrected.

4. In Materials and methods: write the methods only and not why a method was used. You can provide the rationale for the experiment in Results section.

Corrected.

5. Line 140-141: Rephrase

Rephrased.

6. Line 144: rephrase ‘2-2 stress factors’

Rephrased.

7. Line 153,154: ‘Quigene’ should be ‘Qiagen’.

Corrected.

8. Lines 207-209: vaguely written; Rephrase

Rephrased.

9. Line 213: constant should be ‘constitutive’.

Corrected.

10. Fig 4 legend: No need to explain in the method in figure legend.

Explanation removed.

---

## [Decision Letter · Decision Letter 1]

2 Feb 2022

The effects of mycobacterial RmlA perturbation on cellular dNTP pool, cell morphology, and replication stress in Mycobacterium smegmatis

PONE-D-21-26930R1

Dear Dr. Tóth,

We’re pleased to inform you that your manuscript has been judged scientifically suitable for publication and will be formally accepted for publication once it meets all outstanding technical requirements.

Kind regards,

Olivier Neyrolles

Section Editor

PLOS ONE

Additional Editor Comments (optional):

Reviewers' comments:

Reviewer's Responses to Questions

**Comments to the Author**

1. If the authors have adequately addressed your comments raised in a previous round of review and you feel that this manuscript is now acceptable for publication, you may indicate that here to bypass the “Comments to the Author” section, enter your conflict of interest statement in the “Confidential to Editor” section, and submit your "Accept" recommendation.

Reviewer #1: All comments have been addressed

Reviewer #2: All comments have been addressed

2. Is the manuscript technically sound, and do the data support the conclusions?

Reviewer #1: Yes

Reviewer #2: Partly

3. Has the statistical analysis been performed appropriately and rigorously? 

Reviewer #1: Yes

Reviewer #2: Yes

4. Have the authors made all data underlying the findings in their manuscript fully available?

Reviewer #1: Yes

Reviewer #2: Yes

5. Is the manuscript presented in an intelligible fashion and written in standard English?

Reviewer #1: Yes

Reviewer #2: Yes

6. Review Comments to the Author

Reviewer #1: (No Response)

Reviewer #2: The authors have addressed most of my concerns an explained if they could not carry out other suggested experiments. Although they have interesting observations, it is not clear how Rml impacts cellular functions especially replication stress and cell architecture.

7. PLOS authors have the option to publish the peer review history of their article (what does this mean?). If published, this will include your full peer review and any attached files.

Reviewer #1: No

Reviewer #2: No

---

## [Editor Report · Acceptance letter]

11 Feb 2022

PONE-D-21-26930R1 

The effects of mycobacterial RmlA perturbation on cellular dNTP pool, cell morphology, and replication stress in *Mycobacterium smegmatis*

Dear Dr. Tóth:

I'm pleased to inform you that your manuscript has been deemed suitable for publication in PLOS ONE. Congratulations! Your manuscript is now with our production department. 

Kind regards, 

on behalf of

Dr. Olivier Neyrolles 

Section Editor

PLOS ONE